# KETG: A KNOWLEDGE ENHANCED TEXT GENERATION FRAMEWORK

## ABSTRACT

Embedding logical knowledge information into text generation is a challenging NLP task. In this paper, we propose a knowledge enhanced text generation (KETG) framework, which incorporates both the knowledge and associated text corpus to address logicality and diversity in text generation. Specifically, we validate our framework on rhetorical text generation from our newly built rhetoric knowledge graph. Experiments show that our framework outperforms baseline models such as Transformer and GPT-2, on rhetorical type control, semantic comprehensibility and diversity.

## 1 INTRODUCTION

Recent pre-trained language models such as GPT-2 can capture clear semantic and syntactic features (Radford, 2018), performing well in machine translation and abstract generation tasks (Li et al., 2016; Wang et al., 2016). However, the application of language models in text generation still needs to be explored. The logic in text generation, especially literature creation, is always obscure, which means they are usually low-frequency, causing the difficulty of modeling by current language models. On the other hand, too much limits of prior information will lead to homogenization in generated texts. To address these issues, (Guan et al., 2020) proposes a knowledge-enhanced pre-training model for commonsense story generation, by transforming the commonsense triples into sentences using a template-based method. However, the template-based transformed sentences from commonsense triples for post-training are rather homogeneous.

In this paper, we introduce an innovative knowledge enhanced text generation (KETG) framework, which incorporates knowledge tuples and their associated sentences in training, such that the logic relation lying in the knowledge tuples can be effectively addressed. Regarding the sentences associated with the knowledge tuples, we may generate the sentences from the tuples by template-based method as in (Guan et al., 2020). However, incorporating real corpus sentences would be more beneficial as they generally exhibit more diversity than those generated from templates, if they are available.

In this way, the generation model can learn the both logicality and diversity in the knowledge tuples and sentences.

We validate our KETG framework on rhetorical text generation, which is an important and essential part in modern literature(Tu et al., 2013).

Rhetoric is quite obscure, requiring strong logical correlation, and a rhetoric knowledge graph with explicit logical information (rather than the commonsense knowledge graph) would be helpful to rhetorical text generation. Unfortunately, to the best of our knowledge, we are not aware of any rhetoric knowledge graph. Hence by using relation extraction methods, we build a rhetoric (specifically, here we refer to metaphor and personification) knowledge graph from a collection of Chinese poems and compositions. With the newly built rhetoric knowledge graph and the corpus from which the knowledge graph is extracted, we train a rhetorical text generation model. Both automatic and manual evaluations show that our KETG model outperforms baseline models on rhetorical type control, semantic comprehensibility and diversity. Experiments also illustrate that incorporating sentences by template-based method in training results in rather similar generated text as the template, while incorporating real corpus sentences brings more diversity in text generation.

To sum up ,the main contributions of this paper are summarized as follows:

1. We propose a KETG framework, which includes both knowledge information and associated sentences in training to address logicality and diversity.

2. We validate our KETG framework on rhetorical (metaphor and personification) text generation. Results show that our KETG framework can generate more reasonable and diverse rhetorical texts, and the rhetoric types can be controlled implicitly.

3. To the best of our knowledge, we build the first Chinese rhetoric (metaphor and personification) graph with 35228 tuples.

## 2 RELATED WORK

**Language Model(LM)**   In order to use as much semantic information as possible, several research work has been conducted. In early stage, researchers focused on the feature-based method to express syntactic and semantic information in texts. However, this kind of method can not solve the problem of polysemy. To improve, (Peters et al., 2018) Elmo is proposed to capture complex word characteristics in texts. Meanwhile, in NLP tasks, massive texts are often unlabeled. To solve this, fine-tuning models are raised, which can learn "common sense" from unlabeled texts. Both Bert and GPT-2 are representative models. (Wang et al., 2019; Ferreira et al., 2019) They have achieved good evaluation results in multiple NLP tasks, such as named entity recognition, Q&A, text classification and text generation.

**Knowledge Enhanced LM**   To mimic human's writing manner, the most basic thing is to ensure that the generated text fluent and semantically understandable. Secondly, the common sense of humankind is also indispensable. Furthermore, aesthetics and logicality make language expressions more vivid, novel and apt. However, it's hard to meet these requirements merely by language models. (Bowman et al., 2015) used common-sense knowledge base in natural language inference(NLI) and NLG. As mentioned in (Zhou et al., 2018), common sense knowledge can promote performance in dialogue generation. (Mihaylov & Frank, 2018) introduced a neural reading comprehension model that encodes external common sense knowledge as key-value memory. (Zhang et al., 2019) introduced a knowledge enhanced pre-trained language framework ERNIE, trying to increase the knowledge representation by masking semantic units such as words and entities. (Guan et al., 2020) proposes a knowledge-enhanced pretraining model for commonsense story generation. They post-train the model on knowledge-augmented data by transforming the commonsense triples into sentences.

**Rhetorical Text Generation**   Rhetoric is an important and essential part in modern literature(Tu et al., 2013). It can express author's passion and grace, improving the aesthetic merit of creations. (Liu et al., 2019) proposed a rhetorically controlled generation model for Chinese poetry generation to govern the rhetorical modes. Through a classifier inserted in the encoder, they can control the rhetorical modes of generated poems. However, it does not include knowledge graph and hence might generate illogical sentences, like "Flakes of snow are flying like snow", which appears to be a metaphor, but includes illogical 'snow like snow'.

## 3 OUR KETG FRAMEWORK

We propose an innovative KETG framework, to combine the knowledge information with text generation models, just like the external device to computer. The architecture could be used to combine different types of knowledge graph with text generation model.

As depicted in Figure 1, we query the keyword in knowledge graph firstly, getting a context vector containing knowledge information. Then, we concatenate the context knowledge vector and the keyword vector, input them together with associated sentence to the language model. In this way, we can highlight the topic in the sentence and potential logical relationship between the entities, forcing the model pay more attention to them. When generating texts, with a given topic word, we get the context knowledge vector in the same way, which then serve as input to the trained model to generate the whole sentence in an auto-regressive manner.

Compared with single topic word, the expanded context knowledge vector can also take the diversity advantage of knowledge graph, make sure the generated sentences full of variety. It's worth

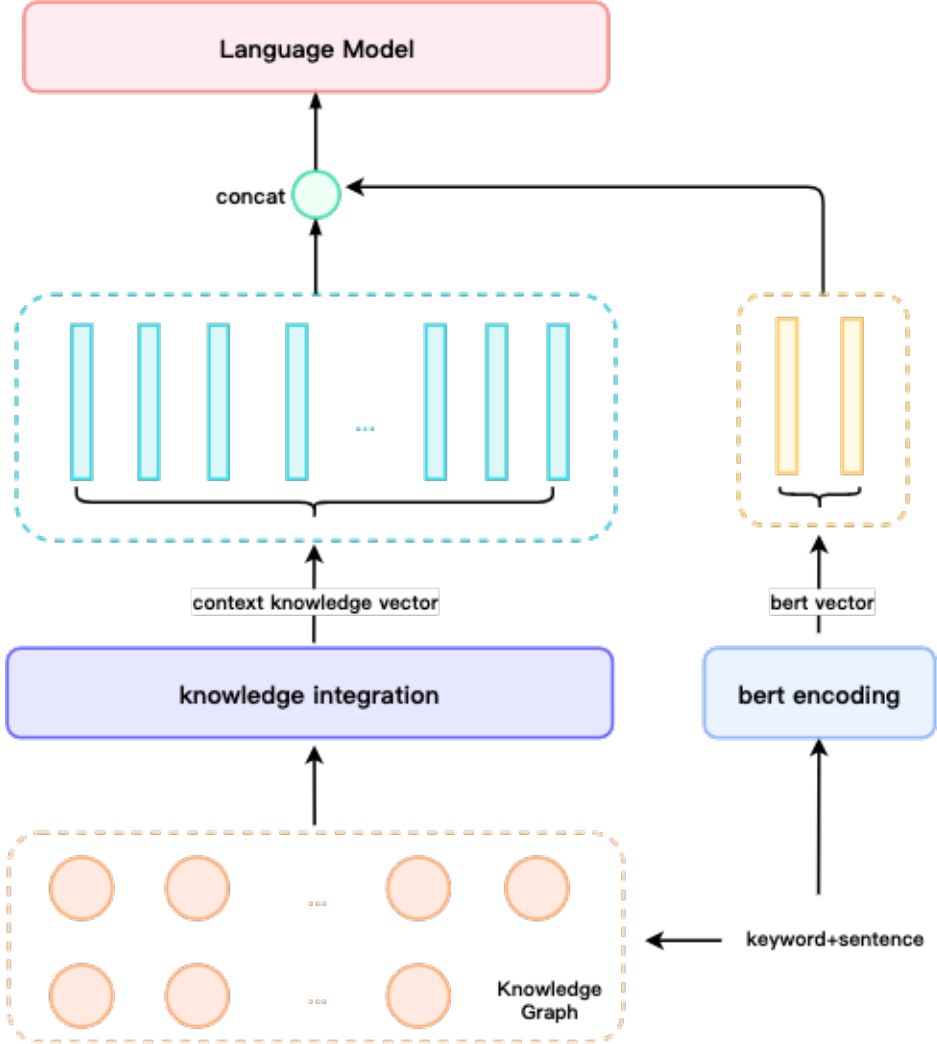

Figure 1: Illustration of our framework.

mentioning that the real corpus sentences are retained in our framework, rather than those generated from templates, which means the generation model can learn the diversity of sentence structure.

In detail, we add [cls] at the beginning of keyword vector and put [mask] to separate them from original sentence. After that, we concatenate them together as the input of text generation model.

Using above approach, we can integrate knowledge information into text generation model naturally. With external knowledge, the generation model can generate more reasonable text, meanwhile captures the significant semantic and syntactic features.

## 4 RHETORICAL TEXT GENERATION

Rhetoric is an essential element in literature. Among 8744 Chinese poems(Liu et al., 2019), 31.4% are metaphor and 18.5% are personification. We also collected 54949 excellent sentences from named composition. Among them, 11989 are metaphor and 28718 are personification. It's obvious that metaphor and personification are the main parts of rhetoric. Therefore we build the rhetorical knowledge graph on metaphor and personification.

### 4.1 RHETORICAL RELATION EXTRACTION

We use relationship extraction algorithm(Rai et al., 2016; Alt et al., 2019) to build our rhetorical graph. Based on bert+crf layer(Huang et al., 2015; Lample et al., 2016; Pramanick et al., 2018), the model is designed to deal with NER(Named Entity Recognition) and relation classification jointly. In addition, we introduces a priori relation graph to filter NER results, which can improve the accuracy of extraction results effectively. Besides, in order to address multiple entities in a sentence, a mechanism called "semi-pointer semi-label" (Su, 2019) is adopted in our model.

### 4.2 CONSTRUCTING RHETORIC GRAPH

We build our rhetorical knowledge graph in three steps.

Firstly, we collect sentences of metaphor and personification from named compositions. Based on coreference resolution rules, we use Stanford Core NLP tools (Manning et al., 2014) to extract metaphor to a tuple of (noumenon, metaphor object, metaphor base), meanwhile personification to a set of (unhuman-subject, human-action/human-emotion). Using the above method, we build a seed rhetorical knowledge graph with 8035 rhetorical sentences, after that manually marked to make sure the accuracy.

Secondly, we trained a rhetorical classifier using this seed graph. We also add 3432 negative examples to prevent over-fitting. The accuracy of the classifier in metaphor is 0.97, while personification is 0.75.

Finally, Based on rules and the above classifier, we continue to expand the data set and retrain the classifier iteratively to build a large rhetorical knowledge graph, including 35228 tuples and 30970 nodes.

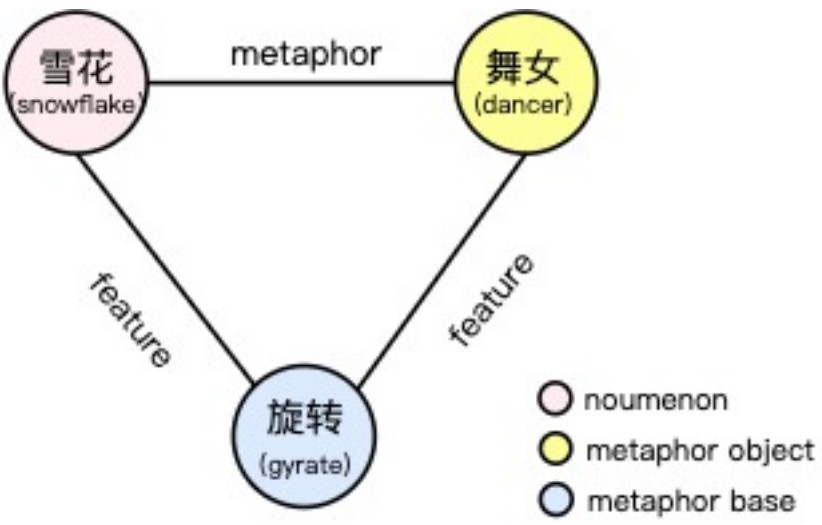

Figure 2: Triangle structure of metaphor.

During the construction, we found that rhetoric relationships have strong logicality, especially metaphor. The common storage mechanism will lead to serious logical errors during query. For example, metaphorical relationships like (snowflake, falling, catkins) and (leaf, falling, snowflakes), will be stored as $[snowflake] - feature - [float] - like - [catkin], [leaf] - feature - [float] - like - [snowflake]$. when searching noumenon "snowflake" in the graph, the result will potentially be $[snowflake] - feature - [float] - like - [snowflake]$.

We design a graph storage mechanism to avoid such illogical problem, the structure is shown in Figure 2. We use a triangle structure to store noumenon, metaphor object and metaphor base. It's worth mentioning that we save the metaphor base as node instead of edge, for that the types of

metaphor base are complicated and varied, saving it as edge will lower search efficiency enormously. Personification is similar to metaphor, but it contains only two entities $[unhuman - subject] - Personification - [human - action/human - emotion]$.

### 4.3 GENERATING WITH RHETORIC GRAPH

Firstly, We query the keyword in rhetorical graph to get a context vector, which including corresponding rhetorical information. For example, in metaphor, the vector contains information of metaphor object and metaphor base.

After, we concatenate the context vector and keyword vector, sending them together with the associated original sentence to text generation model, training the model using the method in Figure 1. During generation, given a topic word and rhetoric type, we get the context knowledge vector in the same way, then generate the corresponding rhetoric sentences in the trained model.

In particular, we use the Top-K generation method, that is, when predicting the next word, we will randomly select one sentence from probable values of top 5. This method can effectively solve the problem of repeated words in the generation.

## 5 EXPERIMENT

### 5.1 DATASET

We build a dataset based on sentences of named compositions, which distributed in various genre and theme. The corpus contains 56814 sentences of three categories. Among them, 17721 sentences are metaphor, 20402 are personification, and the rests are no rhetoric.

Table 1: Automated evaluation results. The best performance is highlighted in **bold**. The PPL score marked with N/A is not comparable, for that the generation approach and dataset of AC model (Liu et al., 2019) are all different from ours. The comparison results are only for reference.

| Model | PPL | Precision | Recall | Rhetoric-F1 |
|---|---|---|---|---|
| GPT2+KG | **7.72** | 0.69 | 0.66 | 0.67 |
| Transformer+KG | 8.69 | **0.74** | **0.73** | **0.73** |
| AC model | N/A | 0.68 | 0.67 | 0.67 |

Table 2: Manual evaluation results. The C indicates Correlation with keywords. F and S denotes the Fluency of generated texts and grammatically comprehensibility. A represents artistic aesthetics.

| Model | C | F | S | A |
|---|---|---|---|---|
| GPT2 | 2.78 | **3.92** | 3.62 | 2.99 |
| GPT2+KG | **3.41** | 3.91 | **3.67** | **3.08** |
| Transformer | 1.97 | 3.14 | 3.02 | 2.91 |
| Transformer+KG | 2.98 | 3.22 | 2.96 | 2.77 |

### 5.2 EXPERIMENT SETTING

In order to demonstrate the effect of knowledge graph, in our experiment, we embed our rhetorical graph to Transformer and GPT-2, and compare the results with vanilla Transformer and GPT-2 models.

The only work on rhetorical text generation we are aware of in literature is in (Liu et al., 2019), which proposes a rhetorically controlled encoder-decoder for modern Chinese poetry generation based on seq2seq framework. They input the rhetorical type into the model explicitly to control the generated rhetorical type. However, without knowledge information, the problem of logic conflict

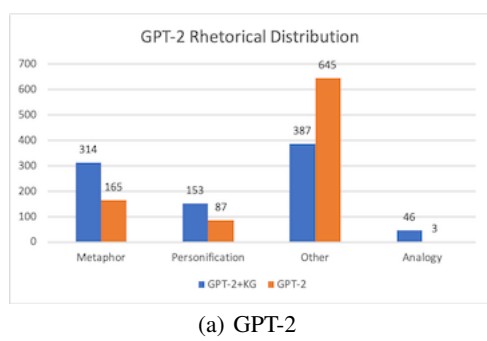 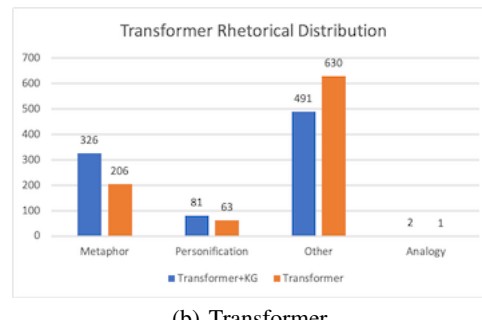

(a) GPT-2                              (b) Transformer

Figure 3: Rhetorical type distribution.The analogy doesn't belong to the rhetorical types studied in this paper, it is manually marked in generated sentences.

is still exist. On the other hand, they generated a poem line by line. That is, when generating i-line, they need to input previous i-1 lines and the topic keywords. However, in our framework, we focus on text generation with topic keyword. Due to the differences of generation approach, we list some experimental results of (Liu et al., 2019) for reference only.

We also carry out an extra experiment, which trains the generation model using template generated sentences(Guan et al., 2020). First of all,we transform the rhetorical triples into template sentences. Then we trained the generation model on templates sentences, instead of real ones. The generated sentences are shown in Table 4. It can be seen that the generated texts are quite homogeneous.

Table 3: Examples of template-based text generation. Examples of template-based transformation of triples in rhetoric knowledge graph. Phrases in **bold** represent the transformed rules.

| Triple | Transformed Sentence | Generative Sentence |
|---|---|---|
| 稻田,海洋,泛起金色的波浪 (paddy,ocean,golden waves) | 稻田**像**泛起金色**的**波浪。 The paddy **is like** golden waves. | 稻谷像泛出黄色的波浪的海洋。 The paddy is like ocean with yellow waves. |
| 路灯,卫兵,站立 (street lamp,guard,stand) | 路灯**像**站立**的**卫兵。 The street lamp **is like** a standing guard. | 路牌像站立的卫兵。 The street sign is like a standing guard. |

### 5.3 AUTOMATIC EVALUATION

**Evaluation Metrics**   We adopt the perplexity score(PPL) and Rhetoric-F1 score to evaluate the generation performance. Perplexity score indicates the fluency of generated sentences. The Rhetoric-F1 score is introduced in (Liu et al., 2019), which can be used to measure the rhetorically controlled accuracy of the generated sentences.

**Evaluation Results**   Our GPT-2+KG obtains a lower perplexity, which suggests that the model is on a par with other models in generating grammatical sentences. When combined with rhetorical knowledge graph, the Rhetoric-F1 of both GPT-2 and Transformer achieve the level of explicit method AC model. It proves that our framework can effectively control the generated rhetorical type implicitly. Detailed results are given in Table 1.

### 5.4 MANUAL EVALUATION

**Evaluation Metrics**   Following previous work(Yi et al., 2018),we consider four criteria for human evaluation:

- Correlation(C): Whether the generated sentence is related to specific keywords;
- Fluency(F): Whether the generated sentence is fluent;

Table 4: Generated examples of different models.

| Triple | Model | Examples |
|---|---|---|
| 大海,雄狮,咆哮 (Sea, lion, roar) | GPT-2+KG | 大海像一头怒吼的雄狮，咆哮着向岸边奔来。 (The sea, like a roaring lion, comes howling toward to the shore.) |
| | Transformer+KG | 大海像一头怒吼的狮子，呼喊着。 (The sea cries out like a roaring lion.) |
| 大海,生气 (Sea, angry) | GPT-2+KG | 这时，我才明白是大海生气了。 (Then I realized that the sea was angry.) |
| | Transformer+KG | 大海呼啸而来。 (The sea is roaring.) |
| 大海 (Sea) | GPT-2+KG | 一天，我和爸爸妈妈去看大海。 (One day, my parents and I went to watch the wave.) |
| | Transformer+KG | 大家都静下来看着自然的海洋。 (Everyone calms down, looking at the natural ocean.) |

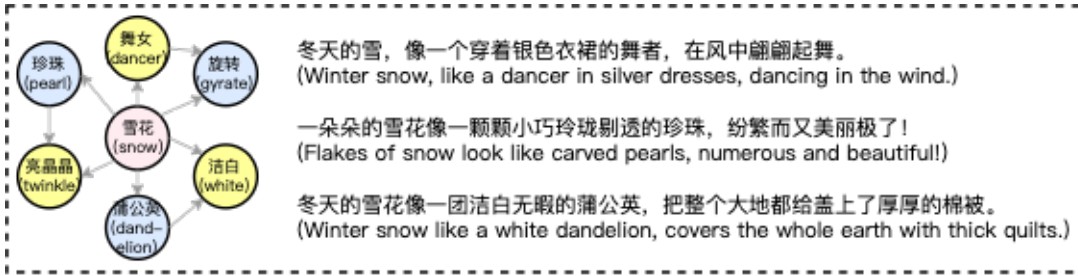

Figure 4: Generated metaphorical sentences of snow by GPT-2+KG.

- Semantic Comprehensibility(S): Whether the generated sentence is comprehensibility in semantics;

- Artistic Aesthetics(A): Whether the generated sentence has artistic beauty;

Each criterion is scored ranging from 1 to 5. We have 180 groups of keywords, each group generates 5 different sentences. We calculate the scores by 5 people voting. That is, removing the highest score and the lowest score, taking the average score of the rest as the final result.

**Evaluation Results** Table 2 shows the results of the human evaluation. In baseline models, it's obviously that GPT-2 performs better than Transformer in all four scores. When combined with rhetorical knowledge graph, the fluency of generated sentences are as good as baseline models, even better for Transformer. On the other hand, the correlation score has improved in both models combined with KG, demonstrating that the knowledge information can be well learned in our framework.

Farther more, when combined with KG, both the semantic comprehensibility and artistic aesthetics of GPT-2 are improved, while the scores of Transformer are slightly reduced. The reason is that the generated sentences of Transformer are always short and simple, too much keywords will limit expression.

In addition, the generated rhetorical type is also manually marked to help us analyze the rhetorical distribution of generated sentences. It can be seen that combined with rhetorical knowledge graph, the number of rhetorical sentences increases obviously. Details can be found in Figure 3.

We also find an attractive phenomenon, that 46 analogy[1] sentences are generated by GPT-2+KG, which can demonstrate the association of a novel internal logic in our graph database. This will help us focus on rhetorical graph inference in the future.

### 5.5 CASE STUDY

In order to further demonstrate how our framework combined with knowledge graph, we display the generated examples in Table 4. It can be seen that our framework learn knowledge information well, control the rhetorical type effectively at the same time. An additional case is shown in Figure 4, illustrating that our framework can take the advantage of knowledge graph, generating diverse texts.

## 6 CONCLUSIONS AND FUTURE WORK

In this paper, we propose a innovative generation framework which can combine knowledge graph with text generation model effectively. In addition, we construct the first Chinese rhetoric graph and devise a graph storage mechanism to resolve the logic conflict problem during query. Experiments show that our method can control the rhetorical types in generated texts, and the texts are more fluent and reasonable at the same time.

Extra experiments show that training with real corpus sentences rather than those generated from templates, generation model can learn the diversity in sentence. However, how to obtain both tuples and associated real corpus, is a restriced condition of our framework.

In future work, it would be very interesting to investigate additional kinds of rhetoric, such as parallelism, to further expand the rhetorical knowledge graph. Meanwhile, we expect to enhance the knowledge inference ability of the knowledge graph by increasing the attributes of nodes.

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
