# OpenReview forum: "KETG: A Knowledge Enhanced Text Generation Framework"
_ICLR.cc/2021/Conference — Reject_

### Official Review · AnonReviewer1 · 2020-10-23
**Official Blind Review #1**

**Rating:** 3
**Confidence:** 4

**Review:**

This paper proposes a framework that embeds rhetoric knowledge into rhetoric text generation to bring diversity in the output sentences, take both knowledge tuples and their associated sentences as model inputs to emphasize logic relations. Besides, this paper builds the first Chinese rhetoric graph with an innovative mechanism of relationship storage.

Pros:
1.	Attempting to address logicality and diversity issues in rhetoric text generation, the author proposes an architecture that incorporates logical knowledge information of different rhetoric entities through a knowledge graph and uses a pre-trained language model as the text encoder.

2.	This paper builds the first Chinese rhetoric (metaphor and personification) graph with 35228 tuples and designed a graph storage method to avoid certain illogical problems.

Cons:
1. The illustration of the framework is ambiguous and confusing.  1.1 How the context vector is calculated by quarrying the keyword in the knowledge graph?  1.2 Will the keywords also act as a part of the input to Bert?

2. AC model cannot serve as the only one baseline method, as the AC model does not use a knowledge graph. It’s unfair to compare two methods with different inputs. Experiments should also include comparisons with other knowledge-enhanced generation methods.

3. The authors claim the strategy yields more diverse syntactic information but not verify it. The paper lacks detailed comparisons on the diversity between the proposed methods and the template-based methods. (For example, a straight-forward template-based method that uses multiple templates can also prompt diversity).

4. The paper hardly presents any technical novelty. The framework is a simple combination of existing methods (pre-trained language models like Bert and GPT2), and the construction of the knowledge graph is not so novel.

5. As this paper proposed a new dataset, authors should make it possible to be followed and reproduced. It’s necessary to release the dataset or elaborate all details about the data source and dataset construction.

Questions:
In the detail of constructing KETG, if the bert_vector is the representation of keyword and sentence, why using [mask] to separate them two instead of using [sep], as [mask] is used to represent the unseen token and [sep] is designed to separate two sentences in the original BERT architecture?

---

### Official Review · AnonReviewer4 · 2020-10-25
**Poorly explained attempt at boosting unconstrained NLG performance**

**Rating:** 2
**Confidence:** 4

**Review:**

The paper is very poorly written. This is in part from English as a second language I suspect, but even assuming some mentorship on the writing, there are too many flaws to consider accepting this paper. The modelling work is so poorly described I can't really comment on it. The majority of my review will just attempt to point out some areas I found lacked clarity.

Some examples of lack of clarity and poor writing:

Introduction:
* "commonsense knowledge graph"; not clear what "commonsense means here.

Related work:
* "fine tuning models are raised". I can only guess at what was meant

 Knowledge enhanced LM section:
* there are several other works looking at incorporating knowledge graph (KG) information into text generation processes, including using popular LM as part of it. What you really mean with this work is something like uncontrolled/unconstrained story generation, although it is never properly defined what your are looking at.

Rhetorical text generation section:
*The negative example "Flakes of snow are flying like snow" -- it isn't certain that such an error is due to not incorporating KG information into the NLG process. This is just a possible hypothesis.

Our KETG framework section:
* "innovative". This is non-scientific. Don't use adjectives to describe your own work.
* "just like the external device to computer". Again, can only really guess at what is meant.
* This whole section is lacking any form of clarity. There are no equations grounding what is precisely done. It isn't even made clear what is assumed by your proposed framework. Where the keywords come from a sentence is not clear. I think this is likely a strong requirement of the approach, but again am guessing given details are lacking. The exact relation to the Guan et al. 2020 paper, which appears to be important, isn't properly described.
* Figure 1: This is very unhelpful. There are so many ways that this figure could be instantiated as an actual model, and no detail is given for how you do so.
* Mentioning [cls] and [mask] without any definition. It can be guessed what you are talking about by readers familiar with BERT and some recent NLP standards, but again the description lacks greatly in clarity.
 Section 4, also called "Rhetorical Text Generation":
* I'm not sure what it means to say a poem is a metaphor. A poem can contain several linguistic traits. How is it determined that a poem is a metaphor?

Constructing Rhetoric Graph section:
* where did the 3422 negative examples come from?
* who manually marked the rhetorical sentences to check accuracy?
* the accuracies of the classifiers are not very good, and yet they are used to bootstrap the rest of the dataset. I assume the data contains a large amount of errors as a result.
* not convinced that "noumenon" is the correct word, but I could be wrong.

Generating with Rhetoric Graph section:
* "sentence to text generation model" ... again, I'm just left guessing at what was actually precisely done.
* "the Top-K", no citation and top-k sampling isn't accurately described.

Remaining sections:
In full disclosure I have only skim read the remaining sections. I was immediately confused by not seeing any results against the proposed KETG name.

---

### Official Review · AnonReviewer3 · 2020-10-29
**Below the bar of ICLR**

**Rating:** 2
**Confidence:** 5

**Review:**

summary:
This paper proposes a framework, KETG, to introduce knowledge during language generation, aiming to enhance the logicality and diversity of generated texts. It is able to utilize both the source sentence and its associated knowledge graph tuples for training.
The authors validate their methods on rhetorical text generation. They first construct a rhetoric graph, then extract context vectors with rhetorical information from it to guide text generation.

pros:
1.	Rhetoric is a stylistic aspect of literature. It is interesting to consider this attribute in natural language generation.

cons:
1.	The writing part of this paper has severe drawbacks.
     a.	The contents are not well-organized at all, the workflow of KETG is still not clear after reading the paper. The overall structure should be modified to present the main ideas clearly.
     b.	Some technical details are missing. In Sec.5, Transformer model and GPT-2 are used as the backbone models and the constructed knowledge graph is embedded into them. But there are not enough details for readers to reproduce the experimental results, e.g. number of epochs and other hyperparameters used during training.
     c.	There are too many typos. Take paragraph 2 in Sec.3 for example.
     line 3: input them together with associated sentence to the ---> input them together with the associated sentence to the;
     line 5: forcing the model pay more attention to them ---> forcing the model to pay more attention to them;
     line6: which then serve as input to the trained model ---> which then serves as input to the trained model

2. This paper aims to introduce knowledge to guide text generation and improve its logicality, but it only evaluates the proposed framework on rhetorical text generation. Actually, there are more representative text generation tasks like dialogue generation, which is also discussed in the related work. More experiments should be included in this work.

Question:
Could you give more details about “knowledge integration” in Fig.1?

---

### Official Review · AnonReviewer2 · 2020-11-04
**Lack of clarity**

**Rating:** 2
**Confidence:** 4

**Review:**

This paper proposes to use a rhetoric knowledge graph for rhetorical text generation. One of its key contributions is to construct a rhetoric knowledge graph by leveraging SOTA NER and relation classification models.  To generate a rhetorical text, the new method starts with sending a keyword to the knowledge graph to retrieve the neighborhood of the keywords as its context words. Both the context words and the original query word are fed into a language model to generate the final word sequence.

Pros:
+ The idea of using a knowledge graph to generate rhetorical text is interesting.

+ There is both qualitative and quantitative analysis of the model performance.

Cons:

- Overall, the paper is poorly written. A significant amount of details are missing such that I do not believe this work is reproducible.

- It is unclear if any datasets are labeled to train NER and relation classification models for knowledge graph construction. What kind of entity types and relations are considered? Which model is actually applied for rhetoric graph construction?

- This work is not compared with the existing SOTA models for poetry generation. It would be more convincing if this work is compared with (Liu et al., 2019).

Questions:

1.  Which transformer model is used as the language model?

2. Which top-K generation method is used for text generation?

3. How do the authors show that the proposed model can eliminate logical inconsistency?

4. Which templates are used for the results in Table 4?

5. What do authors make sure the evaluation of artistic aesthetics is consistent and meaningful?

---

### Decision · Program_Chairs · 2021-01-07
**Final Decision**

**Decision:**

Reject

**Comment:**

While the paper studies an interesting and important problem, namely the language generation, it is poorly written, which makes it difficult to judge its value. The reviewers also expressed concern over the scope of the evaluation and the lack of comparison to SOTA.